# Review of the Health, Welfare and Care Workforce in Tasmania, Australia: 2011–2016

**DOI:** 10.3390/ijerph18137014

**Published:** 2021-06-30

**Authors:** Belinda Jessup, Tony Barnett, Kehinde Obamiro, Merylin Cross, Edwin Mseke

**Affiliations:** Centre for Rural Health, College of Health and Medicine, University of Tasmania, Locked Bag 1322, Launceston, TAS 7250, Australia; tony.barnett@utas.edu.au (T.B.); kehinde.obamiro@utas.edu.au (K.O.); merylin.cross@utas.edu.au (M.C.); edwin.mseke@utas.edu.au (E.M.)

**Keywords:** carers, health workforce, rural health, Tasmania, welfare

## Abstract

Background: On a per capita basis, rural communities are underserviced by health professionals when compared to metropolitan areas of Australia. However, most studies evaluating health workforce focus on discrete professional groups rather than the collective contribution of the range of health, care and welfare workers within communities. The objective of this study was therefore to illustrate a novel approach for evaluating the broader composition of the health, welfare and care (HWC) workforce in Tasmania, Australia, and its potential to inform the delivery of healthcare services within rural communities. Methods: Census data (2011 and 2016) were obtained for all workers involved in health, welfare and care service provision in Tasmania and in each statistical level 4 area (SA4) of the state. Workers were grouped into seven categories: medicine, nursing, allied health, dentistry and oral health, health-other, welfare and carers. Data were aggregated for each category to obtain total headcount, total full time equivalent (FTE) positions and total annual hours of service per capita, with changes observed over the five-year period. Results: All categories of the Tasmanian HWC workforce except welfare grew between 2011 and 2016. While this growth occurred in all SA4 regions across the state, the HWC workforce remained maldistributed, with more annual hours of service per capita provided in the Hobart area. Although the HWC workforce remained highly feminised, a move toward gender balance was observed in some categories, including medicine, dentistry and oral health, and carers. The HWC workforce also saw an increase in part-time workers across all categories. Conclusions: Adopting a broad approach to health workforce planning can better reflect the reality of healthcare service delivery. For underserviced rural communities, recognising the diverse range of workers who can contribute to the provision of health, welfare and care services offers the opportunity to realise existing workforce capacity and explore how ‘total care’ may be delivered by different combinations of health, welfare and care workers.

## 1. Introduction

Despite efforts to more evenly distribute health professionals across Australia, rural and remote areas continue to face unequal access to healthcare services in comparison to their metropolitan counterparts [1]. For rural communities to optimise healthcare service delivery, it is essential that they have clear information on what types of health professionals can be found in their local community, and their capacity to provide healthcare services. Health workforce planning should provide such information, and yet traditional approaches have typically focused on a supply–demand model, with assessments made between the anticipated number of professionals needed within a community and the actual supply [2,3]. Whilst this approach has helped identify potential gaps in service provision and the need for training pipelines to be established, it has also fostered a siloed view of the healthcare workforce. An approach in which significant importance is placed on the availability and accessibility of specific types of health professionals to the neglect of the broader spectrum of healthcare service workforce [4]. For rural communities, there is merit in moving beyond the potential contribution of individual health professionals as currently categorised to establishing an integrated, multidisciplinary framework that focuses more on the services needed within a community and how best to provide those services with the skill mix available in the existing healthcare workforce [4,5,6,7]. This aspirational framework recognises the potential untapped capacity of the broader healthcare workforce to provide services that are needed through mechanisms such as role substitution and role expansion [4,6,7]. This may be useful in situations where certain health professionals are in short supply locally, or the size of the population may not be sufficient to warrant, or be able to afford a particular service.

For rural and remote communities, there is hence the need in health workforce planning to move beyond headcounts of healthcare professionals to a more inclusive assessment of the healthcare workforce. However, a review of the literature reveals that previous evaluations of the healthcare workforce have used traditional approaches, focusing on either specific professions [8,9,10,11] or professional groupings such as medicine [12,13], nursing [13,14], allied health [15,16,17] or oral health [5,18]. The aim of this study was therefore to illustrate a novel approach to health workforce evaluation for rural communities that broadened the scope of workers to better reflect the reality of healthcare service delivery. Specifically, this involved examining the number, type and location of workers involved in the provision of health, welfare and care (HWC) services in Tasmania, Australia between 2011 and 2016.

Tasmania is an island state with a population a little over 540,000 persons dispersed over a geographical area of 68,401 km^2^. Being a largely rural state, Tasmania continues its historical struggle to adequately resource the healthcare sector, with a reliance on attracting skilled professionals from interstate and overseas [19]. Understanding the local HWC workforce more broadly, including the identification of possible service gaps and potential untapped workforce capacity, therefore provides data conducive to effecting innovative and sustainable health workforce policy and planning across the state.

## 2. Materials and Methods

Data on the size and distribution of the Tasmanian population and the Tasmanian HWC workforce was extracted from the 2011 and 2016 Census of Population and Housing conducted by the Australian Bureau of Statistics (ABS). The Census of Population and Housing was considered the most appropriate data source for the purpose of this study because it includes information on regulated and unregulated workers employed across HWC roles, unlike other health workforce data sources (e.g., National Health Workforce Data Set (NHWDS) and Medical Education and Training (MET)) which limit the reporting of data to regulated health professionals.

To create a customised ABS dataset for analysis, comprehensive review of the Australian and New Zealand Standard Classification of Occupations (ANZSCO) was initially undertaken to identify all workers coded in the Census of Population and Housing as providing HWC services. As illustrated by the example of the breakdown of a Major Occupation Group seen in Figure 1, ANZSCO uses a series of numerical codes in a five-tiered hierarchical structure to classify occupations, beginning with one-digit Major Group codes through to six-digit Occupation codes [20]. Following a review of ANZSCO, First Edition, Revision 1 used in the 2011 census [21], 202 six-digit Occupation codes were identified as HWC workers. A review of the updated ANZSCO, Version 1.2 used in the 2016 census [20], subsequently produced 207 six-digit Occupation codes that could be identified as HWC workers. A customised dataset was then created by the ABS detailing the head counts and average hours worked for all three-digit Minor Groups and four-digit Unit Groups that included at least one of the identified six-digit Occupation codes by region, by gender and by level of employment participation (i.e., full-time or part-time) for people who specified their workplace location as Tasmania in the 2011 and 2016 Census of Population and Housing datasets.

To facilitate data analysis, cleaning and reorganisation of the dataset was necessary given that: (a) the ABS had provided additional data for six-digit Occupation codes not included in this study; and (b) ANZSCO codes are grouped according to likeness in training and indicative skill level and not necessarily disciplinary similarity [20]. First, all six-digit Occupation codes not identified as HWC workers were removed from the dataset. To more accurately reflect the contemporary HWC workforce, the remaining three-digit Minor Group codes, four-digit Unit Group codes and six-digit Occupation codes were then regrouped into one of seven different categories based on the type of service they provide: (a) medicine; (b) nursing and midwifery; (c) allied health; (d) dentistry and oral health; (e) health-other; (f) welfare; and (g) carers (Appendix A). Health-other was included as a category to capture those workers who were deemed to contribute to the provision of HWC services but who did not fit under any of the six alternative categories (e.g., biomedical engineer and medical laboratory scientist). In some cases, this regrouping required ANZSCO codes from different Minor Groups and Unit Groups to be clustered together; for example, 4114 Enrolled and Mothercraft Nursing was grouped with 254 Midwifery and Nursing to form the nursing and midwifery category. ABS data custodians advised that given the potential for randomisation of small cell values to either zero or three to ensure that identifiable information is not released, where possible, ANZSCO data should be analysed at the highest hierarchical level possible. Therefore, for this study, we chose to use Minor Group or Unit Group codes in situations where all subordinate four digit or six-digit ANZSCO codes were included. In situations where some ANZSCO codes that comprised a Minor Group or Unit Group were relocated or excluded, then the total of each remaining individual six-digit Occupation code was used instead.

For each category, data were analysed to establish: (a) total headcount, which was an aggregate of all individual headcounts from each Minor Group, Unit Group or individual Occupation codes (Appendix A); (b) total number of FTE positions, which was calculated by summing the total weekly working hours for each included ANZSCO code divided by the average number of hours worked in a full-time job, which for the purposes of this study aligned with the Australian Institute of Health and Welfare’s standard of 40 h per week for medical professionals and 38 h for all other workers [22]; and (c) annual hours of service per capita, which was calculated by dividing the total annual hours of service for each included ANZSCO code by the Tasmanian population figure. Data for each category were analysed for the state and for ABS statistical area level 4 boundaries (SA4) (Hobart, South East, Launceston and North East, and West and North West) (Figure 2). Data for each category were also analysed according to gender as well as employment participation, with full-time employment defined as >35 h per week and part-time employment <35 h per week [23]. If workers were employed but were absent from the workplace for the week preceding census night, they were recorded as ‘away-from-work’ [23]. Finally, dependency ratios were calculated for statistical area level 4 boundaries, which reflected those aged under 15 years and over 65 years as a proportion of the population aged between 15 and 64 years in that region [24]. 

## 3. Results

### 3.1. Tasmanian HWC Workforce

The headcount of the Tasmanian HWC workforce grew by 12.9% between 2011 and 2016, which corresponded to a 10.8% increase in FTE positions and a 7.6% growth in annual hours of service per capita (Table 1). The largest proportional growth occurred in the categories of carers, allied health and medicine, with the annual hours of service provided per capita increasing by 14.4%, 13.1% and 11.4%, respectively. The only HWC category to decline over the five-year period was welfare. Despite a 0.6% increase in headcount, there was a decrease in both the number of FTE positions (−3.4%) and the annual hours of service per capita (−6.2%) provided by this category.

#### 3.1.1. Size

The nursing and midwifery category was the largest single group of workers at both census points, with 6413 people employed in 2011, rising to 6955 in 2016 (Table 1). However, the nursing and midwifery category showed the smallest proportional growth over the five-year period, with an 8.5% increase in headcount, 8.0% growth in FTE positions and a 4.9% increase in annual hours of service per capita. Whilst the category of carers was marginally smaller in size with 4391 people in 2011 and 5278 people in 2016, this category showed the largest proportional growth over the same time. The number of people working as carers rose by 20.2% between 2011 and 2016, which was associated with a 17.8% increase in FTE positions and 14.4% increase in annual hours of service per capita. The smallest category of the HWC workforce was dentistry and oral health, with 720 workers across the state in 2011 and 831 in 2016. Despite being the smallest, the dentistry and oral health category recorded growth between 2011 and 2016, with a 15.4% increase in headcount, 11.7% rise in FTE positions and an 8.5% increase in annual hours of service per capita.

#### 3.1.2. Distribution

When analysed according to SA4, the Hobart region recorded the largest number of Tasmanian residents at both census points (211,656 and 222,356, respectively) and the lowest dependency ratios (52.0% and 55.8%, respectively) (Table 2). The Hobart region also recorded the highest annual hours of service per capita, with 82.3 h in 2011 rising to 90.6 h in 2016. Conversely, the South East recorded the fewest Tasmanian residents at both census points (35,797 and 37,119, respectively) and the least annual hours of service per capita, with 19.9 h in 2011 and 21.3 h in 2016. Whilst the West and North West was the only region to record a decline in population over the five-year period, it recorded an increase in annual hours of service per capita, rising from 59.7 h in 2011 to 68.6 h in 2016. This region also recorded the state’s highest dependency ratio of 58.2% in 2011, rising to 62.9% in 2016. 

Whilst Hobart recorded more annual hours of service per capita at both census points, greater proportional increases in annual hours of service per capita were observed in the Launceston and the North East, and the West and North West regions (14.7% and 15.0%, respectively) over the five-year period. When broken down by category, this greater proportional increase was most evident in nursing and midwifery, where growth in the Launceston and North East (21.7%) and the West and North West (10.7%) regions exceeded that in Hobart (1.5%). Stronger growth was also evident for medicine (21.4%) and dentistry and oral health (18.8%) in Launceston and the North East, whereas allied health (18.4%) growth was strongest in the West and North West. The West and North West was also the only region of the state to record growth in the welfare category, with a 3.8% increase in the annual hours of service per capita over the five-year period. Further, the West and North West was the only SA4 region to record a greater proportional increase in the combined annual hours of service per capita provided by the non-medical workforce (16.1%) when compared to medicine (6.2%). This trend was reversed in all other regions. 

Although the general pattern across the state was of a greater increase in headcount when compared to FTE positions, this was reversed in some HWC categories and in some regions. In particular, nursing and midwifery showed a greater increase in FTE positions when compared to headcount in both the Launceston and the North East (24.3% vs. 23.5%) and West and North West (10.5% vs. 8.4%) regions (Table 3), suggesting an increase in the average hours of work undertaken by nurses over the five-year period. This trend was also observed in the West and North West region for health-other workers, with a 14.0% increase in headcount associated with a 17.9% increase in FTE positions. There was a decline in workers across all categories who reported no fixed working address over the five-year-period. It is likely that the introduction of online census completion in 2016 facilitated the capture of more specific work addresses for some workers (Table 3).

#### 3.1.3. Gender Composition

The overall gender composition of the HWC workforce remained relatively stable over the five-year period, with 75.9% female workers in 2011 and 76.1% in 2016 (Table 4). This gender skew was observed across all categories of the HWC workforce except for medicine, where a greater proportion of the headcount was male at both census points, though the number of females entering medical practice increased from 39.6% to 43.2% over the same period. A similar trend was evident for dentistry and oral health, with a 4.0% decrease in the proportion of males and a 4.6% increase in the number of females. A small shift toward gender balance was also observed in the female dominated carers category, with the number of male care workers increasing from 18.8% in 2011 to 20.6% in 2016.

#### 3.1.4. Employment Participation

Over the five-year period there was a shift toward part-time work, with the percentage of the HWC headcount reporting part-time employment rising from 45.7% in 2011 to 48.7% in 2016 (Table 5). The data also showed a greater proportional increase in FTE positions filled by part-time HWC workers (22.2%) when compared to full-time HWC workers (5.5%) over the five-year period. Despite this overall shift toward part-time work across the HWC workforce, some categories (medicine, allied health, welfare, and dentistry and oral health) maintained a higher proportion of full-time workers, whereas carers and nursing and midwifery demonstrated a predominately part-time workforce. Of note, gender difference was evident in the growth of the part-time HWC workforce, with a greater increase in the number of males working part-time (24.1%) compared to females (19.6%) over the five-year period (Table 5). However, after reviewing each of the individual categories comprising the HWC workforce, only carers had more males employed part-time than full-time, with 460 part-time workers in 2011 rising to 625 in 2016, compared to 313 full-time workers in 2011 rising to 399 in 2016. More female workers were found to be employed part-time compared to full-time in the categories of carers, nursing and midwifery, health-other and welfare (Table 5).

The number of workers away-from-work increased by 9.3% for the total HWC workforce between 2011 and 2016 (Table 5). However, this was underscored by an increased number of females away-from-work (13.6%) and fewer males away-from-work (−2.3%) over the five-year period. All categories of the HWC showed more female workers away-from-work in 2016 except nursing and midwifery (−2.2%), with the largest increases observed in dentistry and oral health (62.5%), carers (30.1%), welfare (25.0%) and allied health (24.7%). In contrast, the number of males away-from-work declined across most categories, with large reductions noted in welfare (−61.0%) and dentistry and oral health (−45.5%). The only categories to record a rise in the number of males away-from-work over the five-year period were health-other (55.1%) and carers (28.3%).

## 4. Discussion

This study has illustrated a novel approach to health workforce analysis that has included all types of HWC workers to better reflect the reality of changes in healthcare service delivery in rural communities. The overall expansion of the Tasmanian HWC workforce between 2011 and 2016 is consistent with national data illustrating strong growth in all sectors of the healthcare industry over the same period [25,26,27]. Using a broad framework that included all types of HWC workers has, however, illustrated the varying growth in categories in response to changing healthcare demands. For Tasmania, growth has been largest in the carers, medicine and allied health categories over the five-year period, with increased demand for these workers possibly attributable to population demographics and policy initiatives. Tasmania has the highest national dependency ratio [24], with almost 1 in 5 people aged over 65 years [28]. With a program of aged care reforms announced in 2011 that included focusing on enabling the elderly to remain at home for longer [29], this may explain the stronger growth for carers observed in this study. The elderly are also the most prolific users of medical care [30], which is a further possible contributing factor to the greater growth in the medicine category. However, the introduction of the National Disability Insurance Scheme (NDIS) may also help explain the observed changes in the Tasmanian HWC. With Tasmania having the highest prevalence of disability in Australia [31], the state was selected as one of four trial sites to introduce the NDIS in July 2013 [32]. The subsequent progressive rollout of the NDIS has likely increased the number of carers and allied health professionals in the state, with both categories of workers in high demand to support NDIS participants [33]. Tasmania also has some of the highest rates of chronic health conditions nationally [34]. Growth in the allied health category may therefore also be associated with the increasing utilisation of allied health services to support chronic disease management [35].

The only category of the Tasmanian HWC that declined was welfare. Despite a small gain in headcount, the overall number of FTE positions and per capita services provided by this category fell over the five-year period. This finding is in contrast to national data indicating solid growth in the collective welfare workforce over the same period [36]. However, some professions comprising the welfare category, such as welfare workers, have declined in headcount over the past five years nationally [37], suggesting that specific roles within this category are changing or may be less attractive. Certainly, the decline in FTE despite growth in headcount observed in this study suggests that there may be fewer full-time jobs or workers may not want to work as many hours in their current roles. However, the observed decline may also reflect a changed job market, where investment has shifted away from welfare roles in favour of other HWC categories. Policies such as the Better Access initiative, introduced in 2006 and then refined in 2009–2010, have likely fostered this shift [38]. Designed to improve the provision of mental health services to the community, the initiative introduced a range of new items to the Medicare Benefits Schedule to better remunerate general practitioners, psychologists, social workers and occupational therapists for their time providing mental health services [38]. With mental health care provided by medical and allied health professionals now more affordable, this may in part explain the declining FTE and per capita welfare services observed in this study [38,39].

By including a broader range of HWC workers, this study moved away from examining the distribution of the healthcare workforce based on the number of registered professionals per 100,000 population [12,13,14,15,18] and computed the annual hours of service per capita provided by individual categories and the total HWC workforce. This approach recognised that populations that are widely distributed, especially rural and remote communities, will rarely attract their notional ‘per capita’ allocation of a particular professional discipline. These circumstances may require other HWC workers to either increase service provision, or alternatively change roles, to compensate for a particular service not being represented locally. By determining both the breadth of the HWC workforce, together with the total hours of service provided to communities, a more holistic estimate of local healthcare service capacity was possible. This could help identify opportunities for role substitution and expansion to build local capability and increase the flexibility of the existing workforce to meet local healthcare needs when the recruitment and retention of some professions may be very difficult. Such circumstances underscore the growing recognition of the health-welfare interface, resulting from major national reforms in hospital services, primary care, aged care and disability support that promote and require integrated cross-sectorial care [40]. However, given that unmet need may be provided by unregulated workers, authorities need to generate a stronger framework for regulation across healthcare more broadly to ensure the quality and safety of services provided [40].

For rural communities such as Tasmania, whose population is widely dispersed across small, geographically distinct localities, the use of this per capita hours of service model would therefore be transformative for local health service managers facing the challenges of uneven distribution of the HWC workforce and service availability across the state. In the SA4 Hobart area, for example, where there are more total hours of service per capita than any other region of the state, health service managers can plan to deliver a range of healthcare services using a varying mix of HWC workers. The Hobart region includes Tasmania’s capital city and is home to just under half of the state’s total population. Some centralisation of services could be expected, particularly specialist medical services that are notably clustered in metropolitan areas across Australia [41]. Although evidence of service decentralisation emerged over the five-year period, with greater proportional increases in annual hours of service per capita in both Launceston and the North East and the West and North West regions compared to the Hobart region, all other SA4 regions of the state will need to consider exactly what types of services they can practically offer from the HWC workforce available. In the West and North West region for example, where long-standing difficulties recruiting medical professionals are expected to persist [42], an increase in all non-medical HWC workers is likely to continue to ensure healthcare is provided to the community. 

Consistent with the profile of the broader Australian HWC workforce [25], this study observed the Tasmanian HWC to be highly feminised, with around three quarters of workers being female. Medicine, dentistry and oral health, allied health, health-other and welfare also appear to be becoming increasingly feminised, with the proportion of females in each of these categories increasing over the five-year period. This mirrors national data [25], confirming the long-standing observation that more females are entering some traditionally male dominated professions. Of note is that female HWC workers typically work fewer hours per week in comparison to their male counterparts [15,29,43,44]. Therefore, as the proportion of female HWC workers continues to grow, so too will recruitment efforts given that additional workers will be required to achieve the meaningful change in FTE needed to keep pace with increasing service demands.

Unlike all other HWC categories, the Tasmanian care workforce showed growth in the number of male care workers over the five-year period which is consistent with national reports illustrating a gradual increase in the number of male care workers in recent years [26]. An important factor behind this rise may be that industries such as agriculture and manufacturing have endured substantial job losses over the past decade [45], resulting in men having to consider alternative employment options. Part of the attraction to care work may be the relatively short commitment to vocational study to achieve a relevant qualification. Further, care roles are typically dominated by part-time work arrangements [26,27], which may appeal to men wishing to balance work and family commitments.

Although some categories of the Tasmanian HWC including medicine, allied health, welfare and dentistry and oral health continue to be dominated by full-time roles, this study observed a clear shift toward part-time employment for all workers comprising the HWC workforce. This trend appears characteristic of the systemic change in workforce participation observed both in Tasmania [46] and nationally across all industries and sectors, not just healthcare [47]. The reasons for the change are underpinned by both supply and demand factors, such as a move toward flexible and more cost-effective job offerings to the personal decision of workers to engage in part-time capacity to balance work commitments with study and family care [47]. Although this personal choice may be thought to centre around those workers with young children, the literature illustrates that older workers nearing retirement [41], as well as workers under the age of 40 years [44], may also shy away from full-time roles. Given that both age cohorts are strongly represented in the Tasmanian population, and hence the Tasmanian HWC, this may account in part for the systemic shift toward part-time work. Future health workforce evaluation will need to capture both headcount and FTE to monitor the impact of declining workforce participation on the ‘total care’ provided to local communities [48].

The present study also found evidence of the ‘disappearing working man’ phenomenon [46,49], with more male workers taking up part-time employment than female workers over the five-year period. This trend challenges the traditional cultural norm of a ‘breadwinning male’ within each family and highlights that flexibility is likely to be equally important in recruiting workers, both male and female, to all health, welfare and care roles. While the move to part-time employment may in some circumstances be by choice, it has also been highlighted that around one quarter of part-time workers would like to work additional hours [47]. Certainly, this study has illustrated the capacity of part-time HWC workers to work additional hours, with a greater percentage increase in FTE compared to headcount of part-time workers across most categories. This may well reflect greater economic rationalisation, with employers’ preference for a casual or part-time workforce so that hours can then flexibly increased or decreased in line with organisational pressures. If this is the case, the move away from secure employment arrangements has substantial implications for attracting HWC workers to the state, who may be reluctant to relocate from interstate for anything less than permanent full-time employment, particularly for those professions who are not supported by a local supply of graduates and are hence reliant on interstate and international migration [19].

While this study has illustrated changes in the Tasmanian HWC, the findings must be interpreted with caution. Namely, census data is obtained through self-report and is not corroborated. Further, headcounts of smaller occupations may also be somewhat inaccurate given the random adjustment of cell values to ensure identifiable information is not released. In an effort to control for this, this study has used aggregated totals for professional groups where possible, thus helping to maintain the integrity of the dataset. Other features of the census also present as inherent limitations to reporting health workforce data, including the ‘away from work’ coding of census respondents. The current dataset observed large numbers of HWC workers recorded on census night as ‘away from work’, and with no way to determine their average hours of work, their contribution to HWC per capita hours of service across the state remains unmeasured. Finally, census data also fails to reflect the time spent on non-clinical versus clinical duties by HWC workers. Should administrative demands have grown over time, then the actual hours of direct clinical care provided to the community has likely been overreported in this study. Despite these limitations, the Census of Population and Housing provides a valuable resource for tracking both regulated and unregulated HWC workers, particularly in the absence of an alternative comprehensive dataset.

## 5. Conclusions

Using a novel approach to health workforce evaluation that broadened the inclusion of all types of health, welfare and care workers has better reflected the reality of healthcare service delivery in Tasmania. For rural communities, such an approach provides the opportunity to recognise the collective contribution of both regulated and unregulated workers to healthcare service provision and the compositional shifts that may result from workforce shortages or in attempting to provide services that cannot be sustained due to small population size. The introduction of a more person-centred approach to calculating healthcare service provision which describes service hours per capita underscores that healthcare is not provided by individual professions, but by the health system as a whole. For rural and remote populations who vary in size and locality, the broad approach to health workforce evaluation and per capita hours of service model used in this study would be transformative, providing local communities with clear evidence of the types of HWC workers already available to them and ‘total care’ accessible. Determination of how best to use available workers and how to deliver this ‘total care’ can then be informed by local population demographics and health needs.

## Figures and Tables

**Figure 1 ijerph-18-07014-f001:**
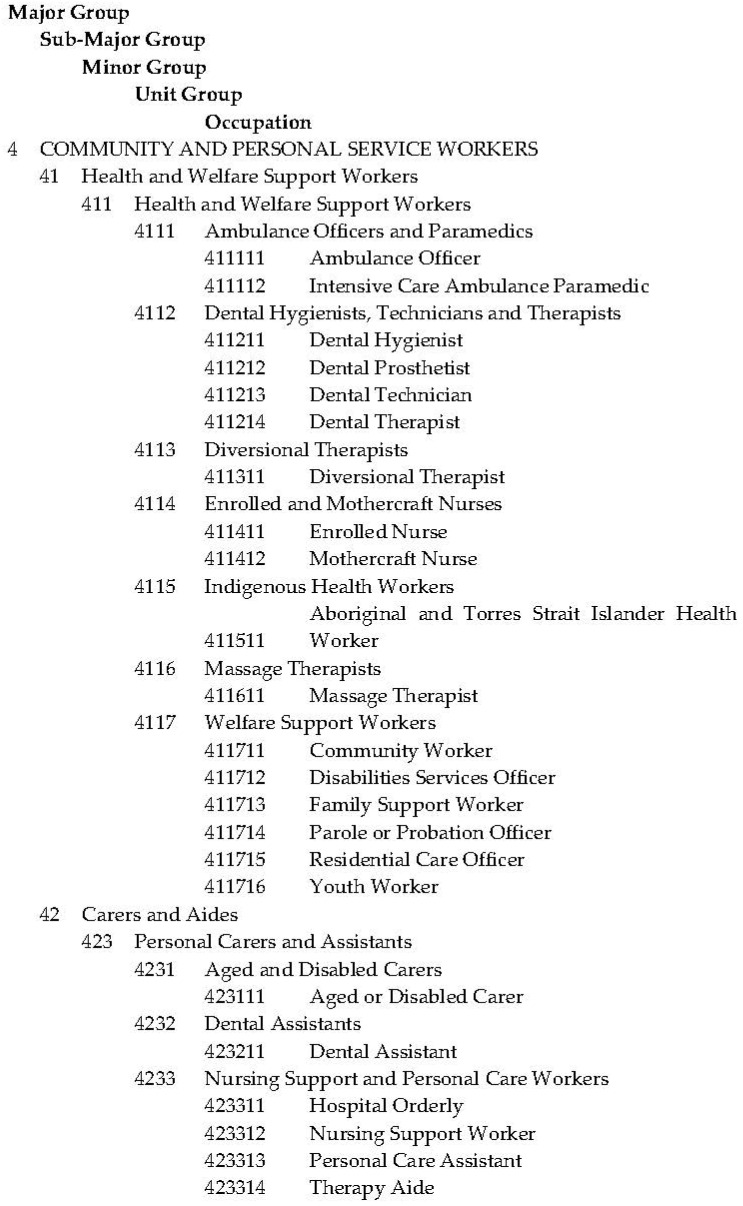
Extract from ANZSCO, Version 1.2.

**Figure 2 ijerph-18-07014-f002:**
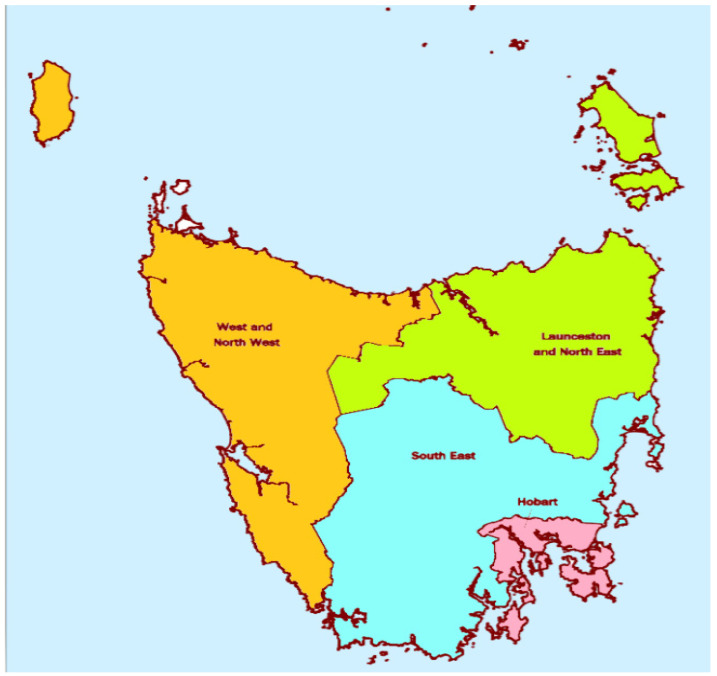
Statistical Areas Level 4-Tasmania.

**Table 1 ijerph-18-07014-t001:** Changes in the Tasmanian health, welfare and care workforce between 2011 and 2016.

	Headcount			Full Time Equivalent			Annual Service Hours per Capita		
	2011	2016	% Change	2011	2016	% Change	2011	2016	% Change
Medicine	1574	1874	19.1	1613.4	1850.6	14.7	6.8	7.5	11.4
Nursing and Midwifery	6413	6955	8.5	5109.8	5518.0	8.0	20.4	21.4	4.9
Allied Health	3050	3627	18.9	2671.6	3110.3	16.4	10.7	12.1	13.1
Dentistry and Oral Health	720	831	15.4	637.1	712.0	11.7	2.5	2.8	8.5
Health-Other	3654	4134	13.1	2994.5	3351.7	11.9	11.9	13.0	8.7
Welfare	2698	2714	0.6	2324.8	2245.5	−3.4	9.3	8.7	−6.2
Carers	4391	5278	20.2	3153.1	3714.1	17.8	12.6	14.4	14.4
Total	22,500	25,413	12.9	18,504.2	20,502.1	10.8	74.2	79.8	7.6

**Table 2 ijerph-18-07014-t002:** Changes in the regional distribution of the Tasmanian health, welfare and care workforce between 2011 and 2016.

SA4 Region	Headcount			Full Time Equivalent			Annual Service Hours per Capita		
	2011	2016	% Change	2011	2016	% Change	2011	2016	% Change
Hobart (Population)							(211,656)	(222,356)	(5.1)
Dependency Ratio							52.0	55.8	3.8
Medicine	881	1082	22.8	878.8	1055.0	20.0	8.6	9.9	14.3
Nursing and Midwifery	3138	3429	9.3	2499.3	2666.0	6.7	23.3	23.7	1.5
Allied Health	1529	1847	20.8	1307.8	1575.6	20.5	12.2	14.0	14.7
Dentistry and Oral Health	355	417	17.5	313.7	363.8	16.0	2.9	3.2	10.4
Health-Other	1764	2144	21.5	1476.5	1761.9	19.3	13.8	15.7	13.6
Welfare	1231	1272	3.3	1056.7	1071.3	1.4	9.9	9.5	−3.5
Carers	1661	2192	32.0	1237.2	1643.0	32.8	11.6	14.6	26.4
Total	10,559	12,383	17.3	8770.0	10,136.5	15.6	82.3	90.6	10.0
South East (Population)							(35,797)	(37,119)	(3.7)
Dependency Ratio							55.6	63.7	8.1
Medicine	25	40	60.0	26.6	33.3	25.4	1.5	1.9	20.9
Nursing and Midwifery	119	129	8.4	94.4	94.8	0.5	5.2	5.0	−3.1
Allied Health	49	63	28.6	36.9	44.0	19.3	2.0	2.3	15.0
Dentistry and Oral Health	3	13	333.3	2.5	11.5	361.3	0.1	0.6	344.9
Health-Other	79	101	27.8	59.2	83.7	41.4	3.3	4.5	36.4
Welfare	74	58	−21.6	57.5	42.9	−25.5	3.2	2.3	−28.1
Carers	116	132	13.8	82.4	88.1	6.9	4.5	4.7	3.1
Total	465	536	15.3	359.4	398.3	10.8	19.9	21.3	6.9
Launceston and North East (Population)							(137,558)	(140,484)	(2.1)
Dependency Ratio							55.3	60.0	4.7
Medicine	378	481	27.2	405.4	502.6	24.0	6.1	7.4	21.4
Nursing and Midwifery	1724	2129	23.5	1383.6	1719.1	24.3	19.9	24.2	21.7
Allied Health	795	992	24.8	712.8	848.5	19.0	10.2	11.9	16.6
Dentistry and Oral Health	188	235	25.0	163.1	197.9	21.3	2.3	2.8	18.8
Health-Other	966	1067	10.5	803.2	856.4	6.6	11.5	12.0	4.4
Welfare	724	726	0.3	622.0	595.7	−4.2	8.9	8.4	−6.2
Carers	988	1268	28.3	691.4	878.9	27.1	9.9	12.4	24.5
Total	5763	6898	19.7	4781.5	5599.2	17.1	69.0	79.1	14.7
West and North West (Population)							(109,152)	(109,024)	(−0.02)
Dependency Ratio							58.2	62.9	4.7
Medicine	227	253	11.5	233.8	247.9	6.0	4.5	4.7	6.2
Nursing and Midwifery	1118	1212	8.4	901.4	996.4	10.5	16.3	18.1	10.7
Allied Health	521	637	22.3	475.8	562.6	18.3	8.6	10.2	18.4
Dentistry and Oral Health	128	161	25.8	116.3	135.3	16.4	2.1	2.5	16.5
Health-Other	623	710	14.0	484.8	571.4	17.9	8.8	10.4	18.0
Welfare	513	564	9.9	461.7	478.9	3.7	8.4	8.7	3.8
Carers	808	1104	36.6	610.0	781.2	28.1	11.0	14.2	28.2
Total	3938	4641	17.9	3283.7	3773.8	14.9	59.7	68.6	15.0
No Fixed Address									
Medicine	53	15	−71.7	54.9	10.7	−80.6			
Nursing and Midwifery	336	59	−82.4	258.5	40.8	−84.2			
Allied Health	182	65	−4.3	140.3	47.5	−66.2			
Dentistry and Oral Health	30	3	−90.0	26.8	3.5	−87.0			
Health-Other	201	95	−52.7	145.5	56.1	−61.5			
Welfare	175	71	−59.4	134.7	43.7	−67.6			
Carers	814	581	−28.6	519.0	327.0	−37.0			
Total	1791	889	−50.4	1279.8	529.2	−58.6			

**Table 3 ijerph-18-07014-t003:** Change in annual service hours per capita by region.

	Hobart			South East			Launceston and the North East			West and North West		
	2011	2016	% Change	2011	2016	% Change	2011	2016	% Change	2011	2016	% Change
Population	211,656	222,356	5.1	35,797	37,119	3.7	137,558	140,484	2.1	109,152	109,024	−0.02
Dependency Ratio	52.0	55.8	3.8	55.6	63.7	8.1	55.3	60.0	4.7	58.2	62.9	4.7
Medicine	8.6	9.9	14.3	1.5	1.9	20.9	6.1	7.4	21.4	4.5	4.7	6.2
Nursing and Midwifery	23.3	23.7	1.5	5.2	5	−3.1	19.9	24.2	21.7	16.3	18.1	10.7
Allied Health	12.2	14	14.7	2	2.3	15	10.2	11.9	16.6	8.6	10.2	18.4
Dentistry and Oral Health	2.9	3.2	10.4	0.1	0.6	344.9	2.3	2.8	18.8	2.1	2.5	16.5
Health-Other	13.8	15.7	13.6	3.3	4.5	36.4	11.5	12	4.4	8.8	10.4	18
Welfare	9.9	9.5	−3.5	3.2	2.3	−28.1	8.9	8.4	−6.2	8.4	8.7	3.8
Carers	11.6	14.6	26.4	4.5	4.7	3.1	9.9	12.4	24.5	11	14.2	28.2
Total	82.3	90.6	10	19.9	21.3	6.9	69	79.1	14.7	59.7	68.6	15

**Table 4 ijerph-18-07014-t004:** Changes in gender composition of the Tasmanian health, welfare and care workforce between 2011 and 2016.

	Headcount (%)			FTE (%)		
	2011	2016	% Change	2011	2016	% Change
Medicine						
Male	949 (60.3)	1072 (57.2)	13.0	1055.8 (65.4)	1157.8 (62.6)	9.7
Female	624 (39.6)	809 (43.2)	29.6	555.4 (34.4)	699.8 (37.8)	26.0
Nursing and Midwifery						
Male	667 (10.4)	734 (10.6)	10.0	625.4 (12.2)	680.7 (12.3)	8.8
Female	5756 (89.8)	6225 (89.5)	8.1	4498.8 (88.0)	4844.7 (87.8)	7.7
Allied Health						
Male	914 (30.0)	1058 (29.2)	15.8	924.7 (34.6)	1041.2 (33.5)	12.6
Female	2156 (70.7)	2567 (70.8)	19.1	1757.1 (65.8)	2063.3 (66.3)	17.4
Dentistry and Oral Health						
Male	194 (26.9)	190 (22.9)	−2.1	193.7 (30.4)	187.8 (26.4)	−3.1
Female	522 (72.5)	641 (77.1)	22.8	439.8 (69.0)	523.9 (73.6)	19.1
Health-Other						
Male	1019 (27.9)	1091 (26.4)	7.1	964.8 (32.2)	1015.0 (30.3)	5.2
Female	2652 (72.6)	3048 (73.7)	14.9	2037.0 (68.0)	2338.1 (69.8)	14.8
Welfare						
Male	902 (33.4)	844 (31.1)	−6.4	860.7 (37.0)	787.5 (35.1)	−8.5
Female	1798 (66.6)	1861 (68.6)	3.5	1466.5 (63.1)	1452.8 (64.7)	−0.9
Carers						
Male	825 (18.8)	1089 (20.6)	32.0	670.6 (21.3)	869.8 (23.4)	29.7
Female	3563 (81.1)	4180 (79.2)	17.3	2474.9 (78.5)	2841.3 (76.5)	14.8
Total						
Male	5470 (24.3)	6078 (23.9)	11.1	5295.7 (28.6)	5739.7 (28.0)	8.4
Female	17,071 (75.9)	19,331 (76.1)	13.2	13,229.5 (71.5)	14,764.0 (72.0)	11.6

**Table 5 ijerph-18-07014-t005:** Changes in employment participation of the Tasmania health, welfare and care workforce between 2011 and 2016.

	Full-Time						Part-Time						Away from Work *		
	Headcount (%)			FTE (%)			Headcount (%)			FTE (%)			Headcount (%)		
	2011	2016	% Change	2011	2016	% Change	2011	2016	% Change	2011	2016	% Change	2011	2016	% Change
**Male**															
Medicine	785	853	8.7	979.3	1042.8	6.5	122	179	46.7	68.6	104.3	51.9	43	39	−9.3
Nursing and Midwifery	403	433	7.4	467.0	488.3	4.6	212	251	18.4	150.0	178.9	19.3	47	46	−2.1
Allied Health	667	778	16.6	786.5	903.7	14.9	188	248	31.9	113.5	147.8	30.3	47	44	−6.4
Dentistry and Oral Health	146	141	−3.4	169.8	159.2	−6.3	44	48	9.1	28.1	32.9	17.1	11	6	−45.5
Health-Other	713	715	0.3	803.5	825.1	2.7	256	309	20.7	148.4	180.5	21.6	49	76	55.1
Welfare	580	526	−9.3	685.5	605.5	−11.7	268	264	−1.5	156.4	153.9	−1.6	59	23	−61.0
Carers	313	399	27.5	375.8	460.9	22.6	460	625	35.9	279.4	386.9	38.5	53	68	28.3
Total	3607	3845	6.6	4267.4	4485.4	5.1	1550	1924	24.1	944.4	1185.2	25.5	309	302	−2.3
**Female**															
Medicine	380	466.0	22.6	444.6	535.9	20.5	196	284	44.9	105.4	158.3	50.3	49	58	18.4
Nursing and Midwifery	2296	2265	−1.4	2599.4	2535.4	−2.5	2919	3435	17.7	1856.7	2263.5	21.9	538	526	−2.2
Allied Health	1100	1264	14.9	1222.6	1385.1	13.3	860	1140	32.6	488.0	671.4	37.6	150	187	24.7
Dentistry and Oral Health	290	340	17.2	305.6	358.2	17.2	211	276	30.8	122.8	164.4	33.8	16	26	62.5
Health-Other	1064	1223	14.9	1213.5	1377.4	13.5	1401	1602	14.3	810.1	923.4	14.0	191	219	14.7
Welfare	875	819	−6.4	977.9	905.4	−7.4	808	917	13.5	466.2	531.9	14.1	100	125	25.0
Carers	954	1061	11.2	1105.5	1222.5	10.6	2329	2783	19.5	1315.5	1564.5	18.9	269	350	30.1
Total	6959	7438	6.9	7869.1	8319.9	5.7	8724	10437	19.6	5164.7	6277.4	21.5	1313	1491	13.6
**Total**															
Medicine	1162 (73.8)	1319 (70.4)	13.5	1420.5 (88.0)	1579.5 (85.4)	11.2	317 (20.1)	464 (24.8)	46.4	173.6 (10.8)	263.3 (14.2)	51.7	94 (6.0)	91 (4.9)	−3.2
Nursing and Midwifery	2699 (42.1)	2697 (38.8)	−0.1	3068.3 (60.0)	3026.2 (54.8)	−1.4	3137 (48.9)	3688 (53.0)	17.6	2009.3 (39.3)	2448.3 (44.4)	21.8	585 (9.1)	567 (8.2)	−3.1
Allied Health	1764 (57.8)	2050 (56.5)	16.2	2014.2 (75.4)	2298.8 (73.9)	14.1	1078 (35.3)	1389 (38.3)	28.8	623.1 (23.3)	823.2 (26.5)	32.1	195 (6.4)	225 (6.2)	15.4
Dentistry and Oral Health	434 (60.3)	474 (57.0)	9.2	472.5 (74.2)	509.7 (71.6)	7.9	255 (35.4)	324 (39.0)	27.1	150.5 (23.6)	197.0 (27.7)	30.9	28 (3.9)	34 (4.1)	21.4
Health-Other	1765 (48.3)	1929 (46.7)	9.3	2016.1 (67.3)	2200.7 (65.7)	9.2	1642 (44.9)	1900 (46.0)	15.7	949.2 (31.7)	1096.8 (32.7)	15.5	242 (6.6)	286 (6.9)	18.2
Welfare	1464 (54.3)	1358 (50.0)	−7.2	1674.2 (72.0)	1533.0 (68.3)	−8.4	1066 (39.5)	1195 (44.0)	12.1	619.7 (26.7)	693.0 (30.9)	11.8	156 (5.8)	163 (6.0)	4.5
Carers	1276 (29.1)	1455 (27.6)	14.0	1493.0 (47.4)	1676.1 (45.1)	12.3	2785 (63.4)	3407 (64.6)	22.3	1588.2 (50.4)	1949.2 (52.5)	22.7	329 (7.5)	414 (7.8)	25.8
Total	10,564 (47.0)	11,282 (44.4)	6.8	12,158.9 (65.7)	12,824.1 (62.5)	5.5	10,280 (45.7)	12,367 (48.7)	20.3	6113.5 (33.0)	7470.7 (36.4)	22.2	1629 (7.2)	1780 (7.0)	9.3

* Away from work denotes people who were on leave or otherwise temporarily absent from the workplace at the time of census counting.

## Data Availability

The data presented in this study are available on request from the corresponding author. The data are not publicly available from the Australian Bureau of Statistics but were obtained through the purchase of customised datasets.

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
