# Peer review of "Review of the Health, Welfare and Care Workforce in Tasmania, Australia: 2011–2016"

_ijerph, 2021, doi:10.3390/ijerph18137014_

Round 1

Reviewer 1 Report

Thank you for this interesting manuscript.  There are two areas in which it might be improved:

  1. You seem to refer to the goal of having identical access to services across a state such as Tasmania.  Given that there are many health-related services that require a certain population density to assure safe and affordable service, not every health service may be available everywhere, especially in remote areas.  The goal in that case would be to assure an organized system of assessment and access, including transportation and telehealth options, which would require some shifting in the composition of the workforce.
  2. Your use of extremely data-dense tables makes great sense for the reader interested in the detailed analysis, but can be overwhelming to the more out-come oriented reader, such as someone looking for potential policy impact.  Give some though to working with an expert on graphics to see if something more reader-inviting can be developed. 

Author Response

Reviewer 1

Thank you for this interesting manuscript.  There are two areas in which it might be improved:

Thank you for the positive feedback and encouragement to continue to improve the manuscript.

You seem to refer to the goal of having identical access to services across a state such as Tasmania.  Given that there are many health-related services that require a certain population density to assure safe and affordable service, not every health service may be available everywhere, especially in remote areas.  The goal in that case would be to assure an organized system of assessment and access, including transportation and telehealth options, which would require some shifting in the composition of the workforce.

Thank you for your comments.  Although it was not the intention of the paper to refer to the goal of identical access to services across Tasmania, we can appreciate how this may have been interpreted from the discussion given that we have stated that there is uneven access to healthcare across the state.

The paper recognises from the outset that rural communities are underserved in various ways, and it is not the intention to argue the traditional line of ‘we need more of x professionals’ in each community to address inequity.  Rather, the aim of the study was to illustrate a novel methodology that may help rural communities evaluate their existing health workforce more broadly to consider the untapped potential for healthcare service delivery.  This is emphasised in the discussion, where we have stated ‘For Tasmania, whose population is widely dispersed across small, geographically distinct localities, the use of this per capita hours of service model would therefore be transformative for local health service managers facing the challenges of uneven distribution of the HWC workforce and service availability across the state.’

Your use of extremely data-dense tables makes great sense for the reader interested in the detailed analysis, but can be overwhelming to the more out-come oriented reader, such as someone looking for potential policy impact.  Give some though to working with an expert on graphics to see if something more reader-inviting can be developed. 

Thank you for your suggestion.  We agree that the tables in the paper are data dense.  However, the tables were carefully reviewed prior to submission and have already been simplified as much as possible.  As you have indicated, there will be readers interested in the detailed analysis and therefore we feel it is important to  continue to include the detail provided, particularly to support the discussion.  As such, reducing the tables further would detract from the paper.  Currently, the formatting only allows the tables to be presented in portrait.  It would be expected that with further formatting in landscape, the readability of these tables would substantially improve.  Should the tables continue to be challenging to embed in the text, then they could be included as a supplementary file instead.  This would seem most logical for Table 5 which is the most data dense of all the tables and most likely to impact on readability.

Reviewer 2 Report

This is a very interesting and well written manuscript. The methodology used is robust and well justified and the results are clear. Simply put, the ground presented by the actors is that i) for rural communities to optimize healthcare service delivery, it is essential that policy-makers have clear information on what types of health professionals can be found in their local community, and their capacity to provide healthcare services; ii) the optimization of healthcare service delivery could be done through role substitution and expansion to build local capability and increase the flexibility of the existing workforce to meet local healthcare needs when the recruitment and retention of some professions may be very difficult.

Although the authors do collect, analyze and provide the data that could be used by policy-makers in Tasmania (in spite of the limitations recognized by the authors at the end of the manuscript), they fail to explain the public policy part, i.e. how role substitution and expansion could occur. By not being clear about this, the manuscript falls short of what was expected. In this sense, it is missing a solid conceptual discussion on what integration in health professions means, as well as a practical discussion on if this could be possible in Australia and according to its health legislation (e.g. on the role of each profession).

On the other hand, the discussion of the results could be improved if the authors tried to link them with Australian health policies: were there any political and public policy changes that can explain some of the results? Of course, this discussion could be more accurate if the time frame of analysis was longer. In this sense, why was the data collected only for 2011 and 2016?

Author Response

Reviewer 2

This is a very interesting and well written manuscript. The methodology used is robust and well justified and the results are clear. Simply put, the ground presented by the actors is that i) for rural communities to optimize healthcare service delivery, it is essential that policy-makers have clear information on what types of health professionals can be found in their local community, and their capacity to provide healthcare services; ii) the optimization of healthcare service delivery could be done through role substitution and expansion to build local capability and increase the flexibility of the existing workforce to meet local healthcare needs when the recruitment and retention of some professions may be very difficult.

Thank you for the positive feedback.

Although the authors do collect, analyze and provide the data that could be used by policy-makers in Tasmania (in spite of the limitations recognized by the authors at the end of the manuscript), they fail to explain the public policy part, i.e. how role substitution and expansion could occur. By not being clear about this, the manuscript falls short of what was expected. In this sense, it is missing a solid conceptual discussion on what integration in health professions means, as well as a practical discussion on if this could be possible in Australia and according to its health legislation (e.g. on the role of each profession).

Thank you for your suggestions.  The overall aim of this paper was to demonstrate a novel approach to workforce planning which broadened the scope to include health, welfare and care workers, all of whom contribute to healthcare service delivery.  Although role substitution and expansion have been suggested as possibilities throughout the paper, it was not the purpose to find evidence of this, but rather to suggest a novel way of interpreting data collected at a national level from the health workforce perspective.  Role expansion and substitution could certainly be at play but we did not specifically seek to investigate this phenomena.  This will be for a future paper!

On the other hand, the discussion of the results could be improved if the authors tried to link them with Australian health policies: were there any political and public policy changes that can explain some of the results? Of course, this discussion could be more accurate if the time frame of analysis was longer. In this sense, why was the data collected only for 2011 and 2016?

Thank you for your suggestions.  The purpose of this paper was not policy focused, but rather to describe a novel methodological approach to workforce analysis that can support rural communities with service provision.  For the first time, a common database with employment patterns of both regulated and unregulated professionals has been analysed simultaneously, illustrating that the ‘health workforce’ is not discrete, but rather spread across multiple ‘workforces’. The discussion has been edited to reflect more succinctly the aims of this study.

The data collected is governed by the Australian census which only occurs every 5 years, which is why the data period is from 2011 and 2016.  The next census is due to occur in 2021.

Round 2

Reviewer 2 Report

I appreciate the authors’ response to my comments. I do understand now better the aim of the article and, because of it, to be quite honest, such a narrow perspective lowers the potential and importance of the study. Still, even bearing in mind such perspective, I reaffirm that the discussion section could be improved if, adding to some of the justifications, the authors tried to link the results with Australian health policies to explain some of the results and the changes occurred between 2011 and 2016.

Author Response

Many thanks for their continued encouragement to revise and improve the submitted manuscript.  The authors have considered your suggestions and have amended the discussion section to include further information regarding Australian health policies that may help to explain the changes seen in the Tasmanian HWC between 2011 and 2016.